# Development and validation of the *Physical Capacity Score* (PiC) to overcome the lack of correlation among traditional physical tests in detecting age-related decline

Gennaro Boccia[1,2☯], Paolo Riccardo Brustio[1,2☯], Anna Mulasso[2,3*], Francesco Tufo[2,3], Alberto Rainoldi[2,3]

1 Department of Clinical and Biological Sciences, University of Turin, Turin, Italy, 2 Neuromuscular Function Research Group, School of Exercise & Sport Sciences, University of Turin, Turin, Italy, 3 Department of Medical Sciences, University of Turin, Turin, Italy

☯ These authors equally contributed to this work.
* anna.mulasso@unito.it

## Abstract

We developed the Physical Capacity Score, a set of validated and commonly used physical tests for adults, administered through a custom-built hardware and software platform that enables automated data collection and analysis. This study aimed to evaluate the platform's repeatability, examine age-related differences, and explore the relationships between different physical capacities in a sample of adults. A total of 812 participants (aged 18–68 years, 63.5% female) were recruited. Participants completed six physical tests: finger tapping, handgrip strength, single-leg stance, sit-and-reach, five-times sit-to-stand, and the YMCA 3-minute step test. Outcome data were standardized by gender (z-scores) and analyzed across age groups using ANOVA. Pearson's correlation coefficient (r) was used to assess redundancy among outcomes, and Principal Component Analysis (PCA) was conducted. In a test-retest analysis, all variables demonstrated coefficient of variation (COV) <10% and intra-class correlation coefficient (ICC) >0.90, except for CoP path length (COV = 10.5%, ICC = 0.64). Correlations among outcomes were weak (r range: 0.036–0.373). While all physical capacities declined with age (p < 0.001), effect sizes varied: from the least to the most age-sensitive outcome ($\eta^2$ values), we found differences in hand-grip strength ($\eta^2 = 0.035$), sit-and-reach ($\eta^2 = 0.050$), finger tapping ($\eta^2 = 0.059$), CoP path length ($\eta^2 = 0.095$), lower-limb power ($\eta^2 = 0.148$), and cardiorespiratory fitness ($\eta^2 = 0.389$). The average PCA component scores revealed large age-related differences ($\eta^2 = 0.301$). Our findings suggest that the developed platform is a valuable tool for assessing physical function, and all physical tests captured distinct aspects of physical capacities. This highlights the necessity of employing a comprehensive battery of tests to gain a holistic understanding of an individual's physical health and detect age-related decline effectively.

**Data availability statement:** All relevant data are within the manuscript and its Supporting Information files.

**Funding:** The author(s) received no specific funding for this work.

**Competing interests:** The authors have declared that no competing interests exist.

## Introduction

Fitness is defined as a set of attributes related to a person's ability to perform physical activities and is determined mostly by a combination of regular activity and genetically inherited ability [1]. Physical fitness enables individuals to carry out daily tasks, to engage in active leisure-time pursuits, and to meet unforeseen emergencies without undue fatigue; for these reasons it is sometimes referred to as dynamic state of energy or "vitality" [2,3]. As individuals age, it becomes increasingly important to comprehensively assess their physical capacities to mirror the multifaceted nature of physical health and to gain a deeper understanding of their longevity prospects. In this regard, it has been highlighted the need for a more comprehensive approach to health assessment, one that considers a wide range of physical capacities rather than just assessing a single one [4,5]. These capacities, including cardiovascular fitness, muscle strength, flexibility, hand dexterity, and balance, can provide valuable insights into an individual's overall physical health and functional status [4]. By measuring these various physical capacities, researchers and healthcare providers can better identify individuals at risk of functional decline, disability, and adverse health outcomes, allowing for timely interventions and targeted strategies to promote successful aging [6].

Cardiorespiratory fitness (CRF) is a critical component of health-related physical fitness, reflecting the capacity of the cardiovascular and respiratory systems for prolonged exercise. Extensive research has established CRF's essential role in cardiovascular health and overall mortality risk, showing a strong inverse relationship between CRF and the risk of hypertension, dyslipidemia, and cardiometabolic conditions [7]. CRF is also a more powerful predictor of mortality than measures like BMI or body fat percentage [8]. Clausen et al found that middle age men with higher CRF have up to 6 years of extended longevity [9] even after adjusting for other risk factors, such as age, smoking, and BMI. CRF is a potentially stronger predictor of mortality than established risk factors such as smoking, hypertension, high cholesterol, and type 2 diabetes mellitus [10]. The addition of CRF to traditional risk factors significantly improves the reclassification of risk for adverse outcomes. For example, pre-operatory CRF predict the rate of cardio-respiratory adverse events after intra-abdominal surgery [11]. CRF is also positively linked to cognitive function. In older adults, higher CRF levels measured 25 years earlier have been associated with better verbal memory and faster psychomotor speed [12].

Muscular strength is another crucial component of overall physical fitness. Muscular strength is defined as the capacity to generate force against a resistance. As it is influenced by various factors, including the size and number of involved muscles, the proportion of muscle fibers activated, the coordination of muscle groups, and osteo articular characteristics there is no single test that can measure the overall muscle strength [1]. Muscle strength enables individuals to perform a wide range of physical tasks, from everyday activities to more demanding athletic pursuits. In older adult, muscle strength is correlated with cognitive capacities [13] as well as mortality higher levels of handgrip strength were associated with a 31% reduced risk of all-cause

mortality compared with lower muscular strength [14]. While handgrip strength has been widely used as a measure of general muscle function, lower limb muscle strength and power may be more functionally relevant, particularly in the context of aging and various clinical conditions. A significant decline in muscle strength is observed in older adults, with the lower body muscles being more affected than the upper body [15]. Muscle power declines at even larger rate [16]. This reduction in lower limb muscle strength and power is associated with balance and mobility issues, leading to physical impairment and loss of independence [17]. Decreased muscle strength in the lower extremities can have profound implications on an individual's ability to perform daily activities, such as walking, climbing stairs, and maintaining balance, which are essential for maintaining autonomy and quality of life [15].

While measures of strength and power focus on musculoskeletal fitness, the assessments of balance and dexterity focus more on sensorimotor function. Test that evaluate an individual's balance is crucial because not only impacts individual's physical independence, but also the risk of falls and other injuries [18]. Identifying individuals at high risk and addressing the underlying factors is essential for preventing falls and maintaining functional ability [18,19]. Hand dexterity, the ability to manipulate objects with precision and control, is a crucial aspect of human functionality that underpins a wide range of everyday tasks and professional endeavors [20,21]. Finger tapping tests, which assess an individual's ability to rapidly tap their fingers, have emerged as a valuable tool for evaluating hand dexterity and its underlying neurological and physiological mechanisms. These tests provide insights into the coordination and fine motor control of the fingers, as well as the overall integrity of the sensorimotor system [22,23].

The inclusion of flexibility as a major component of physical fitness is currently under discussion. In his viewpoint, Nuzzo argued that flexibility has little correlation with health and performance outcomes, especially when compared to other components of fitness like cardiovascular fitness or muscle strength [2]. There are conflicting information regarding both the relationship between flexibility interventions and functional outcomes or daily functioning [24]. Nevertheless, training interventions based on stretching, which is the most common method to increase flexibility, has shown robust cardiovascular, metabolic and skeletal muscle adaptations [25]. Similarly to what happens for strength assessment there is no one single test that can measure overall flexibility. Despite a series of stretching exercises targeting the major muscle-tendon units should be tested to have a complete assessment, the most popular stretching test remains the sit and reach test [26], with its intrinsic limitations in terms of validity [27].

For the purpose of evaluate health and functional abilities across the lifespan, the National Institutes of Health developed a specific toolbox (i.e., the NIH Toolbox) which is a comprehensive assessment battery designed to evaluate key facets of an individual's health. The physical capacities domain of the NIH Toolbox assesses an individual's motor function, sensory abilities, and physical fitness [28]. This domain includes standardized measures of strength, dexterity, balance, locomotion, and endurance – all crucial elements of overall physical health and functionality [28–30]. Importantly, the physical capacities domain complements the cognitive and emotional domain, providing a more holistic evaluation of individual's health and wellbeing [28]. Ultimately, the NIH Toolbox's physical capacities domain provides a comprehensive evaluation of persons' physical health and functioning, informing healthcare decision-making and treatment planning [30]. With these information about individual's strengths and weaknesses across the various physical capacities, clinicians can develop targeted intervention strategies to optimize the individual's health and wellbeing [31].

Starting from the idea of the motor function of the NIH Toolbox, we developed a stand-alone platform that adopts a similarly comprehensive approach but is designed for interactive delivery. The purpose-built hardware and software system enable automated and efficient data collection, real-time analysis ensuring accurate and consistent measurements. The aim of this study was to present and test the features of the platform, in particular: i) to evaluate the reliability of the platform in assessing physical capacities; ii) to investigate age-related differences and provide easy-to-understand feedback on the participants' strengths and weaknesses in the different physical capacities; iii) to explore the relationship between different physical capacities in order to understand whether all the selected tests are necessary.

## Materials and methods

### Participants

The subject population for this cross-sectional study was derived from two research projects carried out at the University of Torino (UniTO): *Wellness@Work for UniTO* (W@W; period March 2020 – February 2022) and *Welfare for the health of UniTO students - Wellness4Students* (W4S; period October 2023 – ongoing), which examined the health-related behaviours and promoted active and healthy lifestyles among the university community. W@W and W4S were aimed, respectively, at all personnel (i.e., administrative staff, professors, lecturers, researchers, fellowships, etc) and at first-year students of the University of Torino. For both projects, an initial invitation to participate was sent to all potential participants via their institutional email address with the link to book the evaluation session. Four weeks later, a further email reminder was sent out. Students also received a message via the University's app.

A total of 865 people (468 from W@W – 54.1%, recruitment took place from October 13, 2020 to November 25, 2021; and 397 from W4S - 45.9%, recruitment took place from October 23, 2023 to June 28, 2024) were included in this study. Subjects with missing data on sociodemographic variables and/or the physical tests were excluded from the analysis (n = 51; 5.9%). The study protocol was approved by the Bioethics Committee of the University of Torino (W@W: protocol number 20290 – January 17, 2020; W4S: protocol number 0574345 – October 18, 2023). All individuals gave written informed consent to participate.

### Repeatability analysis

The test–retest analysis was conducted in a convenience subsample (n = 24), in line with common practice for preliminary reliability assessments. Participants were healthy adults (15 males and 9 females, mean age = 36 ± 12 years; body mass = 71.00 ± 10.67 kg; height 1.74 ± 0.08 m; body mass index 23.47 ± 2.50 kg·m$^{-2}$) and were recruited only for analysis of repeatability. Five of them were left-handed. The tests were conducted in two sessions: a pre-test and a post-test, spaced one week apart to ensure consistency and minimize potential learning or adaptation effects. This interval allowed for an adequate assessment of test consistency and repeatability without significant time-dependent variations in participant performance.

### Platform

We developed a platform, comprising hardware and software, called the Physical Capacity (PiC) Score, which enables the automated assessment of physical function using a set of validated and commonly used physical tests for adults aged 18–65 years, including Finger Tapping Test, Hand Grip Strength Test, One Leg Stance Test, Sit and Reach Test, Five Times Sit to Stand Test, and YMCA 3-minute Step Test. The tests are administered using a purpose-built hardware and software platform that enables automated data collection and analysis (Fig 1). The platform dimensions are 1.80 × 1.80 × 1.60 m; handrails are provided to avoid falls. The platform is equipped with a touch-screen computer and a set of embedded sensors to automate test administration and measurement. Sensors position is shown in Fig 1B and described later in the signal analysis section. All the signals were collected and amplified (if necessary) by an electronic board (OT Bioelettronica, Turin, Italy) with a sampling frequency of 200 Hz. The board was connected to the computer via a USB cable and both recording and analysis were performed with custom written code in MATLAB (2022a, MathWorks) using the APP designer interface. While the platform allowed self-administration of the tests, an expert with a Master degree in Sports Science supervised the process to further explain and ensure the correct execution.

### Procedures

Participants stepped onto the platform in front of a screen and, after logging in with an anonymous code, were invited to complete a series of physical tests (see below for the description). A visual and audio guide assisted participants in

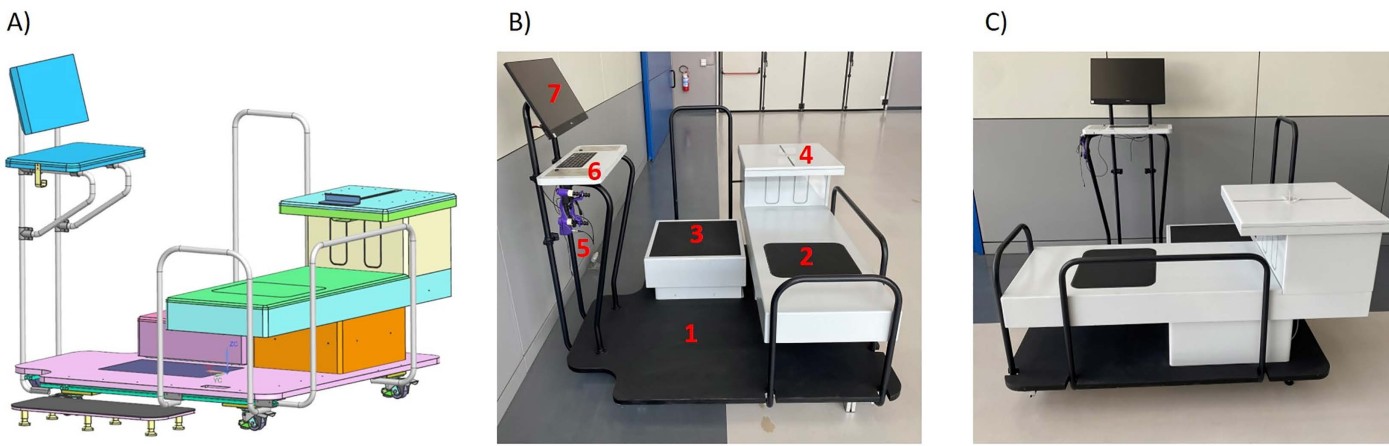

**Fig 1. The initial project (A) and the actual realisation (B and C) of the platform for the assessment of physical capacity (*Physical Capacity score*, PiC).** Panel B shows the embedded sensors: 1) force plate; 2) seat with load cell; 3) step; 4) embedded ultrasound sensor to detect the distance reached during the sit and reach test; 5) handgrip; 6) board with embedded standard keyboard (in black); two custom-made metal keyboards with 1 button each to perform the finger-tap test and (not shown) infrared ear-lobe heart rate detector with 1 m cable; 7) touch screen computer.

performing the tests. The test sequence was fixed and was chosen so that the most demanding tests (i.e., Five Times Sit to Stand Test and YMCA 3-minute Step Test) were performed at the end of the sequence. A video recording of each representative test performance was available to the participant by clicking on a specific virtual button on the screen. In fact, participants had to click on virtual buttons to start each test. After each test, the results of the raw data were displayed on the screen. The whole procedure took an average time of 15 minutes to be completed.

The following tests were included. 1) The Finger Tapping Test is a measure of manual dexterity and motor speed [32]. Participant has to tap a button with the index finger as many times as possible in 10 s. With the palm of the hand resting on the table, the test is performed with both the dominant and non-dominant hands. The outcome measured is the number of taps. 2) The Handgrip Strength Test is the maximum force expressed around a dynamometer (Biometrics Ltd, standard Jamar configuration with five adjustable handles). It is a quick and easy way of providing information about a person's general muscular strength [33]. The test is carried out in a standing position with the shoulder adducted, the arm straight at the side, and the elbow extended. The test was performed with both hands starting with the preferred hand. One repetition is performed for each hand. The outcome measured is the force expressed in N. 3) The Single Leg Stance Test is a measure of static balance [34]. Subjects are instructed to maintain balance on their chosen limb for up to 45 seconds, as calmly and quietly as possible. The other limb is raised with the foot is not in contact with the other ankle) and the arms are crossed on the chest. The test is performed on a force plate surface recording the path of the center of pressure (CoP) for both limbs and without shoes. The outcome measured was the path length of CoP displacement (mm) divided by stature height (cm). 4) The Sit and Reach Test is a measure of hamstring flexibility [26]. Subjects are instructed to sit with the leg straight and the feet against the box, keeping the knees in a locked position. While exhaling, subjects reach forward and slide the hands along the board as far as possible, holding the maximum position for 2 seconds. The outcome measured was the distance covered (cm): higher scores indicate better performance. 5) The Five Times Sit to Stand Test is an expression of lower limb strength [35] and power [36,37]. The test begins in a sitting position, with the feet hip-width apart and the arms crossed on the chest. Subject is instructed to stand up from a chair for five times as fast as possible. When getting up from the chair, the knees are fully extended, and when sitting, the weight is fully leaned back in the chair. The time starts from the auditory "Go" command given by the software and stops when the subject reaches the standing position for the fifth repetition. The outcome measured is the estimated lower limb muscle power (W, see signal analysis

below). 6) The YMCA 3-minute Step Test is a measure of CRF [38,39]. Subjects are asked to step up and down on a 30 cm box for 3 minutes at a step rate of 24 steps/minute. Metronome is set to 96 beats/minute, a step cycle consists of 4 beats. At the end of the 3 minutes, the subject must stop immediately and sit down (within 5 s) to relax. Heart rate (beats per minute, bpm) is monitored and recorded after one minute of recovery. The outcome measured is the estimated maximal oxygen uptake (VO$_2$max).

## Signal analysis

For the Finger Tapping Test, the number of tapping events was determined by a straightforward process of counting the number of zeros or ones during the 10-s trial, as the signal recorded from the button was converted into a binary code, with zeros representing the button being pressed down and ones representing the button being released. For the Hand-grip Strength Test, the maximum strength was defined as the peak force averaged over a 200 ms time window. While performing the test the participants were able to see the displayed level of force produced in order to maximise their effort [2]. Taking into account the Single Leg Stance Test, the CoP trajectory was recorded using the force platform (P6000, BTS, Italy), which track the point of application of the ground reaction forces resultant under the feet [40]. From the 45 s of recorded CoP data, the CoP data were filtered using a 4th order low-pass Butterworth filter with a cutoff frequency of 10 Hz [41] and the CoP path length was calculated [41]. Due to the variability of foot positioning on the force platform, we chose to center the CoP trajectories with respect to their arithmetic mean in our definitions and analysis, in line with previous studies [42]. For the Sit and Reach Test, the distance reached by the participants' hands was measured via a non-contact distance measurement ultrasound module (Ultrasonic Ranger, Grove system, Seed Studio, Shenzen, China) embedded under the board. The distance of the soles of feet was set to 23 cm as per standard procedure [26]. With regard to the Five Times Sit to Stand Test, the participants were seated over the instrumented seat and they placed their feet over the force plate. The signals coming from four load cells placed under the four angles of the seat was summed to obtain a single force signal. Regarding the force plate, for this test only the four vertical components were considered, and they were summed to obtain a signal of ground reaction force (GRF). During the 2 s before starting, weight's subjects was measured with the load cells under the seat. To avoid cheating, the proper execution of the task was checked by measuring the weight of each participant during the seating phase. If participants gave less load to the chair than that measured during the resting phase, a "stop" signal was provided and they were asked to start the test again. The number of repetitions was counted via the force signals of the instrumented seat and the end of the task was calculated by the force plate signals. To do that we firstly individuated the peak GRF corresponding to the last repetition, which was indicated by the fifth lift off from the seat. The end of the last repetition was individuated as when GRF reaches body weight after decreasing and increasing again after peak GRF as previously suggested [43,44]. Muscle power generated when the body's centre of mass moved vertically during the concentric (upward) phase of the sit-to-stand task was assessed using a previously validated equation [36, 37]:

$$Sit\ to\ stand\ power = \frac{Body\ mass \times 0.9 \times g \times [Body\ height \times 0.5{-}chair\ height]}{\frac{Time}{n\ of\ reps} \times 0.5}$$

where body mass is given in kg, g is gravity (i.e., 9.81 m/s$^2$), body height and chair height are given in m, time is given in s, n is the number of repetitions equal to 5, and the other values, i.e., 0.9 and 0.5, are biomechanically derived coefficients [36,37]. Relative sit-to-stand power (W/kg) was then calculated as the sit-to-stand power divided by body mass. For the YMCA 3-minute Step Test, the proper correctness of stepping rhythm was checked via GRF: each time the GRF surpassed the body weight was counted as one step down. A visual and audio cue was given when necessary to inform the participants to increase or decrease the stepping frequency during the test. After 3 minutes a stop signal was given, and the participants were asked to sit immediately. The HR was measured continuously in the first minute of recovery with an ear lobe sensor (Ear-clip Heart Rate Sensor, Grove system, Seed Studio, Shenzen, China). The reference HR have been

averaged in the 5-s around 60s of recovery to improve robustness estimates. Maximal oxygen uptake (VO$_2$max) was then estimated based on previously validated equations [45, 46] to provide a measure of CRF.

## Statistical analysis

**Repeatability.** Pearson's correlation coefficient was adopted to assess the relationship and the consistency between the pre- and post-test scores. Correlation magnitudes were categorized as follows: small (0–0.3), medium (0.3–0.5), and large (>0.5) in accordance with the criteria established by Cohen et al. [47]. According to Cohen [47], eta squared ($\eta^2$) values greater than 0.01, 0.06, and 0.14 correspond to small, medium, and large effect sizes, respectively.

The intraclass correlation coefficient (ICC) for single fixed raters was also utilized to assess the repeatability between pre-test and post-test scores, incorporating both random and systematic errors [48,49]. According to [50], ICC values were interpreted as follows: greater than 0.90 indicates excellent repeatability, 0.75 to 0.90 denotes good repeatability, 0.50 to 0.75 signifies moderate repeatability, and less than 0.50 reflects poor repeatability. Coefficients of variation (COV) were calculated in conjunction with the ICC, with values less than 10% being considered acceptable. Furthermore, Bland-Altman plots [51] with 95% confidence intervals (mean difference ± 1.96 standard deviations) were constructed to examine the systematic variation between the pre-test and post-test data. This approach provides a comprehensive overview of the trends and the variability both across subjects and between tests.

**Descriptive statistics.** Physical tests that involved the performance of one task at a time (e.g., finger tapping, handgrip strength, single leg balance), were averaged to create a single mean value for each variable. Six age groups were considered: 18–20, 21–30, 31–40, 41–50, 51–60, and >60 years of age. Outliers were defined as values exceeding 3 standard deviations (SD) from the sample mean and were removed on a case-by-case basis. Descriptive statistics, including the mean and 95% confidence intervals (CI), were calculated for all age groups.

**Age-related differences.** All data were standardized by gender to obtain the z-scores. The z score of the single leg stance test was multiplied by −1 to obtain a value that increases at increased performance as for the other outcomes. Then, analysis of variance (ANOVA) was conducted for each test to investigate differences between age groups. Finally, a composite score composed by the average of the z-scores of the six tests was compared between age groups with an ANOVA to verify if it could detect age-related differences. The effect size were interpreted as follows [47]: $\eta^2 < 0.01$: negligible; $0.01 \leq \eta^2 < 0.06$: small; $0.06 \leq \eta^2 < 0.14$: medium; $\eta^2 \geq 0.14$: large. Post-hoc pairwise comparisons with Tukey correction were adopted considering the 18–20 group as reference.

**Principal component analysis.** First, the relationships between the test outcomes were analyzed using Pearson's correlation coefficient to detect if there were any redundant outcome. A principal component analysis (PCA) was then conducted to ascertain the factor structure of the whole dataset. To account for potential gender-related differences, all data were standardized by gender. A PCA with oblique rotation (i.e., Promax) was conducted on the covariance matrix to reduce the dimensionality of the data and eliminate redundant variables. The variables and factors were selected based on the following criteria: factor loadings greater than 0.30 and eigenvalues greater than 1.0. The suitability of the correlation matrix for PCA was evaluated through the application of the Kaiser-Meyer-Olkin (KMO) test and Bartlett's test of sphericity, thereby ensuring that the correlation matrix was not an identity matrix.

To obtain a single measure of fitness a Physical Capacity (PiC) Score was calculated. All test outcomes were first standardized by sex (z-scores), PCA was conducted on these standardized variables, and the PiC Score was computed as the unweighted average of the PCA component scores. No weighting factors were applied to individual components.

All statistical analyses were conducted using custom scripts written in R (R version 4.2.3) and MATLAB (2024), with a significance level set at $p < 0.05$.

## Results

### Repeatability

Table 1 reported the analysis of repeatability of the main outcomes while the Bland-Altman plots are reported in the S1 Appendix. The tests re-test analysis performed on 24 participants showed excellent repeatability.

### Age-related differences of individual tests

The mean age of the participants was $35 \pm 15$ years (range 18–68 years) and 63.5% (n = 515) were female. The mean value of the BMI was $22.8 \pm 3.7$ kg/m². The Table 2 shows the outcomes (means and 95%CI) of the physical tests according to gender and age-group.

ANOVA performed on standardized scores showed that all outcomes decreased with age but with different effect sizes. From the least to the most age-sensitive outcome (based on $\eta^2$ size), we found differences in: handgrip strength (F = 5.8, $p < 0.001$, $\eta^2 = 0.035$), sit and reach (F = 8.4, $p < 0.001$, $\eta^2 = 0.050$), finger tapping (F = 10.0, $p < 0.001$, $\eta^2 = 0.059$), single-leg

**Table 1. Consistency, repeatability, and coefficient of variation (COV) for each physical capacity test.**

|  | r (95%CI) | ICC (95% CI) | COV |
|---|---|---|---|
| **Finger tapping** | 0.91 (0.80, 0.96) | 0.92 (0.82, 0.97) | 3.1±2.9 |
| **Handgrip Strength** | 0.96 (0.92, 0.99) | 0.96 (0.91, 0.98) | 4.2±3.5 |
| **Single-leg stance** | 0.72 (0.44, 0.87) | 0.64 (0.33, 0.83) | 10.5±10.0 |
| **Sit and reach** | 0.97 (0.92, 0.99) | 0.97 (0.92, 0.99) | 8.2±10.3 |
| **Lower-Limb Power** | 0.91 (0.80, 0.96) | 0.90 (0.78, 0.95) | 4.7±4.6 |
| **Cardiorespiratory Fitness** | 0.90 (0.79, 0.96) | 0.95 (0.88, 0.98) | 2.1±1.5 |

**Table 2. Descriptive analysis of physical capacity test according to age group.**

|  | Males | | | | | | Females | | | | | |
|---|---|---|---|---|---|---|---|---|---|---|---|---|
|  | <20 yrs | 21-30 yrs | 31-40 yrs | 41-50 yrs | 51-60 yrs | >60 yrs | <20 yrs | 21-30 yrs | 31-40 yrs | 41-50 yrs | 51-60 yrs | >60 yrs |
| **Finger tapping (taps)** | 56.6 (55.6, 57.6) | 56.0 (55.0, 57.1) | 57.5 (55.7, 59.3) | 57.5 (56.2, 58.8) | 55.5 (53.9, 57) | 52.9 (50.0, 55.9) | 54.2 (53.3, 55.0) | 54.4 (53.4, 55.3) | 55.5 (54.1, 56.9) | 53.5 (52.6, 54.4) | 52 (51.1, 52.9) | 47.8 (45.8, 49.8) |
| **Handgrip Strength (N)** | 40.9 (39.6, 42.1) | 43.8 (42.5, 45.1) | 41.1 (39.0, 43.3) | 43.6 (42.0, 45.2) | 40.3 (38.4, 42.1) | 40.4 (36.8, 44) | 26.7 (25.6, 27.7) | 26.8 (25.6, 27.9) | 27.2 (25.5, 28.9) | 28.0 (26.9, 29.1) | 25.7 (24.6, 26.8) | 23.2 (20.8, 25.2) |
| **Single-leg stance (mm/cm)** | 7.6 (7.0, 8.2) | 8.0 (7.4, 8.6) | 8.1 (7.1, 9.0) | 9.2 (8.5, 9.9) | 11.1 (10.2, 11.9) | 13.6 (12.0, 15.2) | 7.1 (6.6, 7.6) | 7.2 (6.7, 7.7) | 7.1 (6.4, 7.9) | 8.0 (7.5, 8.5) | 8.5 (8.0, 9.0) | 10.1 (9.0, 11.2) |
| **Sit and reach (cm)** | 25.6 (23.8, 27.5) | 24.0 (22.1, 26) | 24.9 (21.7, 28.1) | 19.9 (17.6, 22.3) | 19.6 (16.9, 22.4) | 31.3 (29.8, 32.9) | 32.1 (30.4, 33.8) | 29.7 (27.2, 32.2) | 29.2 (27.5, 30.8) | 26.9 (25.3, 28.5) | 25.6 (23.8, 27.5) | 24.0 (22.1, 26.0) |
| **Lower-Limb Power (W/kg)** | 6.5 (6.3, 6.7) | 6.5 (6.3, 6.7) | 6.0 (5.6, 6.3) | 6.0 (5.8, 6.3) | 5.5 (5.2, 5.8) | 5.6 (5.0, 6.2) | 5.6 (5.5, 5.8) | 5.5 (5.3, 5.7) | 5.2 (5.0, 5.5) | 5.1 (4.9, 5.5) | 4.5 (4.3, 4.7) | 3.9 (3.5, 4.3) |
| **Cardiorespiratory Fitness (mL·kg⁻¹·min⁻¹)** | 54.4 (53.4, 55.3) | 53.0 (52.0, 54.0) | 51.4 (49.7, 53.0) | 47.9 (46.6, 49.1) | 46.2 (44.7, 47.7) | 46.8 (44, 49.7) | 48.1 (47.3, 48.9) | 47.6 (46.7, 48.5) | 46.5 (45.1, 47.8) | 42.6 (41.7, 43.5) | 38.5 (37.6, 39.3) | 34.6 (32.7, 36.5) |

Data are reported as mean and 95% confidence intervals.

balance (F = 16.8, $p < 0.001$, $\eta^2 = 0.095$), sit-to-stand power (F = 28.1, $p < 0.001$, $\eta^2 = 0.148$), and CRF (F = 102.9, $p < 0.001$, $\eta^2 = 0.389$). Post hoc outcomes are reported in the Fig 2. Briefly, adopting as a reference the 18–20 years of age, the first age-group that showed a statistically significant difference was the 51–60 for finger tapping, 41–50 for single-leg stance and sit and reach, 21–30 for sit-to-stand power and CRF (Fig 2). Handgrip showed the highest value for the 41–50 while the 61–65 group was not different from the 18–20.

## Correlation

As shown in Fig 3, the Pearson correlation analysis revealed mostly weak associations among physical capacity variables. Briefly, the Handgrip strength showed a medium positive correlation with sit-to-stand power (r = 0.328; $p < 0.01$) and a small positive correlation with Sit and reach (r = 0.127; $p < 0.01$), and finger tapping (r = 0.137; $p < 0.01$). CRF showed a medium correlation with sit-to-stand power (r = 0.373; $p < 0.01$), a small correlation with Sit and reach (r = 0.249; $p < 0.01$) and single-leg stance (r = 0.254; $p < 0.01$). Furthermore, sit-to-stand power also showed a small positive correlation with finger tapping (r = 0.128, $p < 0.01$) and sit and reach (r = 0.186; $p < 0.01$).

## Principal component analysis

With regard to the PCA, the overall Kaiser-Meyer-Olkin measure was 0.533, and Bartlett's test of sphericity was statistically significant ($\chi^2 = 393.4$; $p < 0.001$), indicating that the data could be decomposed into factors. The items and factors were selected according to the criteria of factor loadings exceeding 0.30 and eigenvalues exceeding 1.0. The PCA revealed three factors, which collectively explained 31.0%, 18.1%, and 16.4% of the total variance, respectively (see Table 3). The total

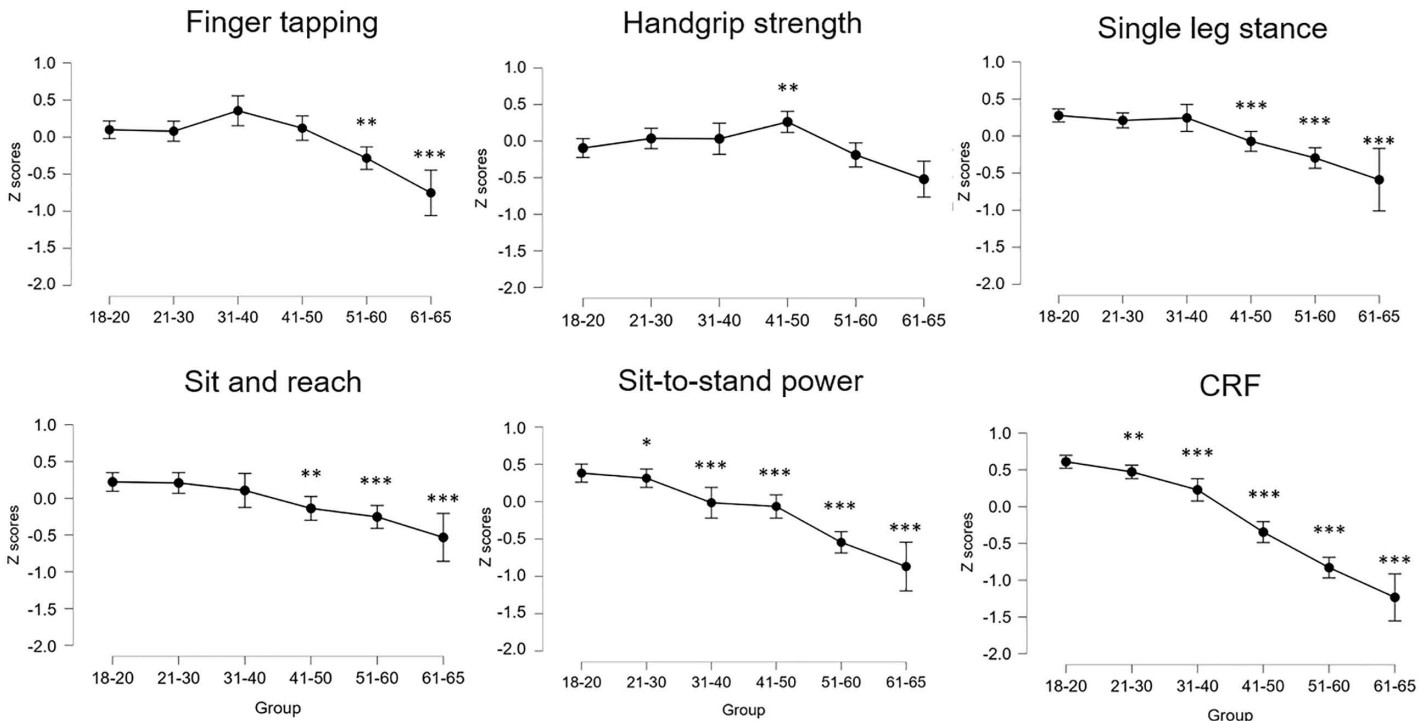

**Fig 2. The outcomes of the six physical tests are reported in standardized units (z scores) with mean and confidence intervals.** Standardization was computed separately for men and women. Post hoc statistically significant differences compared to the 18-20 age group are reported as follows: * $p < 0.05$; **$p < 0.01$; ***$p < 0.001$. CRF, Cardiorespiratory fitness.

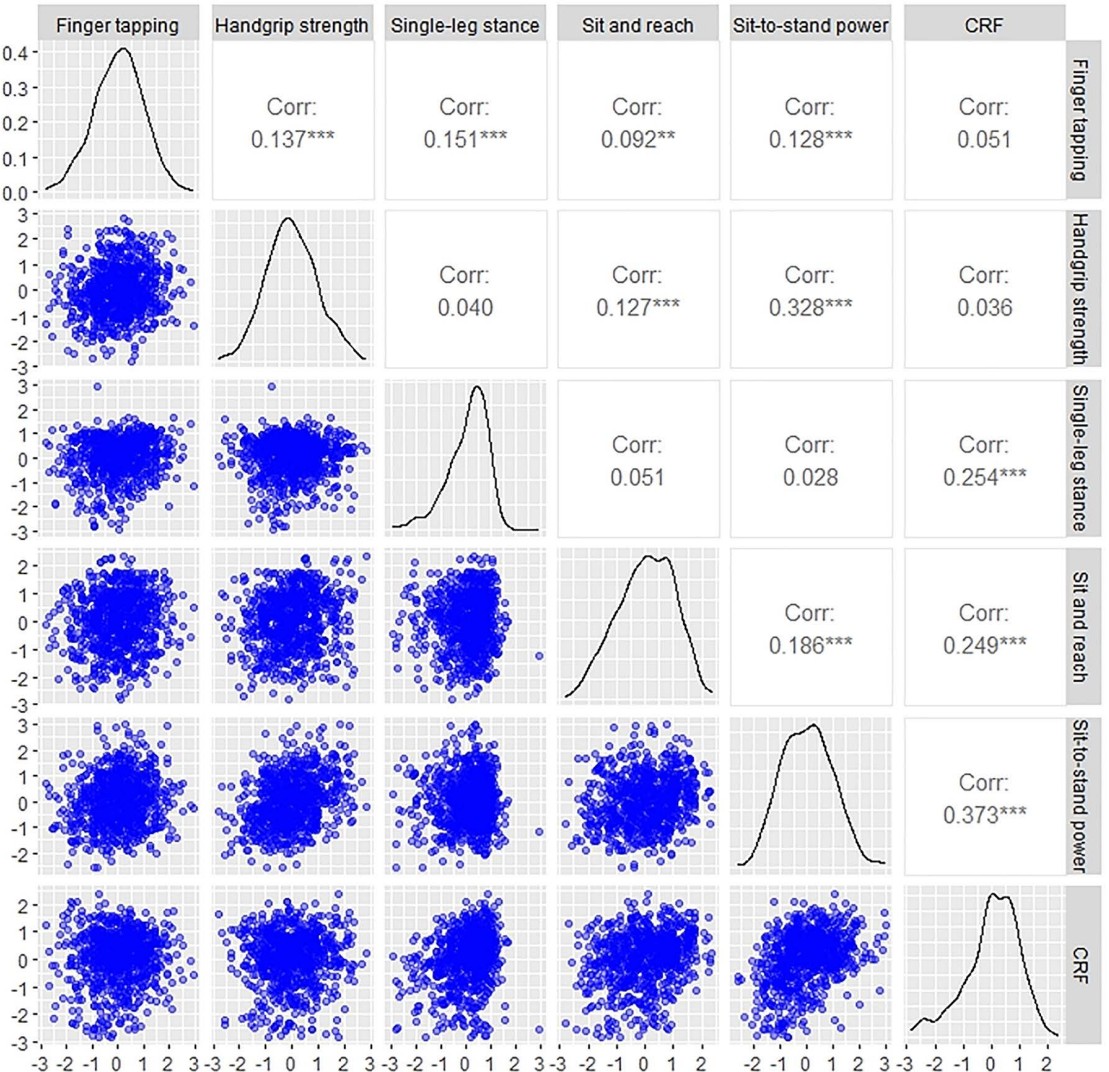

**Fig 3. Scatter plot of z-scores and Pearson's r correlation coefficients are shown for the six physical tests considering the whole sample of 812 participants.**

**Table 3. Principal component analysis.**

|  | Component 1 "Cardiorespiratory Fitness" | Component 2 "Strength-Power" | Component 3 "Sensorimotor" |
|---|---|---|---|
| **Cardiorespiratory Fitness** | 0.831 |  |  |
| **Sit and reach** | 0.657 |  |  |
| **Handgrip Strength** |  | 0.874 |  |
| **Lower-Limb Power** |  | 0.615 |  |
| **Finger tapping** |  |  | 0.925 |
| **Single-leg stance** |  |  | 0.307 |

percentage of variance explained was 65.5%. The initial component designated "Cardiorespiratory Fitness" pertains to the estimated VO$_2$max through the YMCA 3-minute step test and sit-and-reach. The second component, "Strength and power," is associated with handgrip strength and sit-to-stand power. The third component, "Sensorimotor function" is linked to finger tapping and single-leg stance.

ANOVA performed on PCA component scores showed that all outcomes decreased with age but with different effect sizes (Fig 4). The first component "Cardiorespiratory Fitness" (F = 98.7, $p < 0.001$, $\eta^2 = 0.379$) exhibited a significant decline, with the 31–40 age group being the first to show a statistically significant difference compared to the 18–20 age group. The second component "Strength and power" (F = 27.4, $p < 0.001$, $\eta^2 = 0.145$) exhibited a large decline, with the 31–40 age group being the first to show a statistically significant difference compared to the 18–20 age group. The third component "Sensorimotor function" (F = 15.5, $p < 0.001$, $\eta^2 = 0.088$) exhibited an even smaller decline, with the 51–60 age group being the first to show a statistically significant difference compared to the 18–20 age group.

The average of PCA component scores showed a large age-related effect (F = 69.6, $p < 0.001$, $\eta^2 = 0.301$, Fig 5). The first age-group that showed a statistically significant difference from the 18–20 group was the 41–50 group.

A visual example of four selected participants, highlighting strengths and weaknesses identified through physical function assessment, is shown in Fig 6.

## Discussion

We have developed a platform, hardware and software, called the Physical Capacity Score (PiC score), which enables the automated performance of commonly used, validated physical tests. These standardized measurements are commonly thought to be related to health, physical function and longevity. The tests re-test analysis performed on 24 participants showed excellent reliability. We administered the test battery to 812 adults and found that all outcomes were weakly correlated between each other (Pearson's r ranging from 0.036 to 0.373). Moreover, we found varying levels of sensitivity of the tests in detecting age-related differences. From the least to most sensitive, we found: handgrip strength; sit and reach; finger tapping; balance; sit-to-stand power; cardiorespiratory fitness. The average of the six tests z scores showed large age-related sensitivity and the first age-group that showed a statistically significant difference from the 18–20 group was the 41–50 group. The PCA revealed three main components: 1) "Cardiorespiratory Fitness", related to the estimated VO$_2$max and sit-and-reach; 2) "Strength and power", related to handgrip strength and sit-to-stand power; 3) "Sensorimotor function", related to finger tapping and single-leg stance.

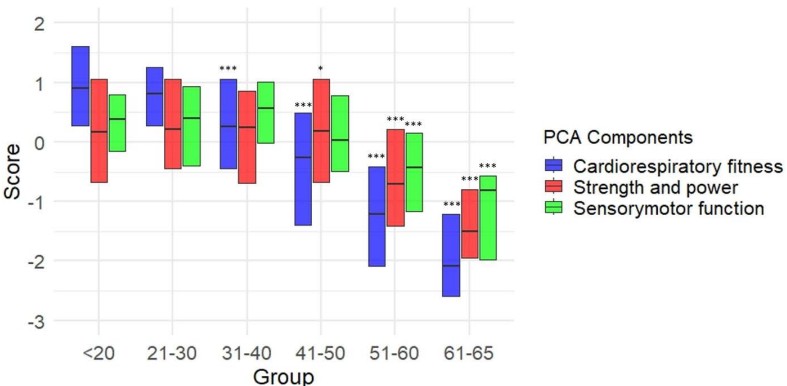

**Fig 4. The median and interquartile range of z-scores obtained in the three components identified by the principal component analysis (PCA) are reported for each age group.** Standardization was computed separately for men and women. Post hoc statistically significant differences compared to the 18-20 age group are reported as follows: * $p < 0.05$; ***$p < 0.001$.

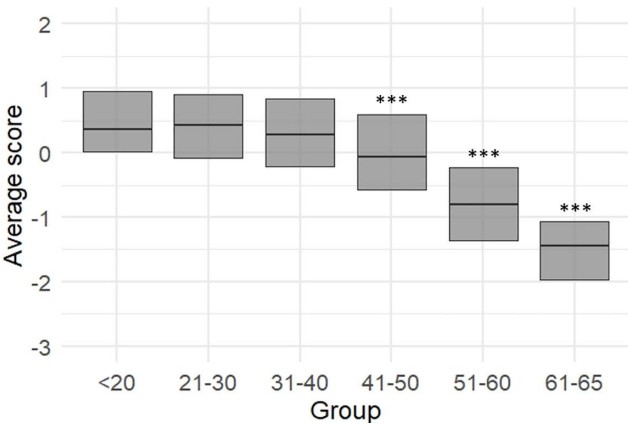

**Fig 5.** **The median and interquartile range of the *Physical Capacity* (PiC) Score, which consists in the average z-scores of the three components identified by the principal component analysis (PCA), are reported for each age group as a measure of overall physical capacity.** Post hoc analysis showed a clear decline in physical performance described by this composite score starting from the age group of 41-50 age group and older with respect to the 18-20 age group. Standardization was computed separately for men and women. Post hoc statistically significant differences compared to the 18-20 age group are reported as follows: ***$p < 0.001$.

**Performance Radar Plot for Selected Subjects**

Finger tapping — Subject A
Handgrip strength — Subject B
CRF — Subject C
— Subject D
Single-leg stance
Sit-to-stand power
Sit and reach

**Fig 6.** **The z-scores for each test are reported for four selected participants that obtained similar average z score (≈ 1.4).** Roughly speaking, the "strengths" and "weaknesses" of each participant can be seen. As the standardization was calculated separately for men and women, the z scores are independent of gender.

The selected tests demonstrated excellent reliability (Table 1) and the ability to discriminate between age groups (Fig 2). We found that the scores derived from these tests were only weakly correlated (Fig 3). These findings suggest that the present set of physical tests may capture distinct aspects of physical capacity and are not redundant. We corroborate the idea that it is necessary to carry out a number of tests to provide a complete assessment of individual physical health [52]. For example, Cooper et al examined the associations between grip strength, sit-to-stand and standing balance (three of the herein adopted tests), with all-cause mortality, finding that a composite including all three measures provided the highest estimate of predictive ability for longevity [53]. These results indicate that the PiC score (Fig 6) provides a

comprehensive assessment of physical function, with the potential to identify individuals at risk of declining health and longevity [54].

Of particular relevance was the lack of correlation between CRF and handgrip strength (r = 0.025). CRF [55] and handgrip strength [14] have been previously demonstrated to be predictive of all-cause mortality. Even though they are both linked to a longer lifespan and to self-rated physical fitness [56], no correlation was observed between these variables in our sample. The correlation between CRF and sit-to-stand power was slightly higher (r = 0.373), but this may be because the two tasks involve the same muscles in a similar way. Both tasks raising the body's centre of mass through leg extension, emphasizing similar muscular demands. In any case, given the null to low correlations between upper and lower limb strength and CRF, the present finding confirms the need to test both physical abilities when a comprehensive health assessment is the goal. We can speculate that concurrent resistance and aerobic training may be an optimal approach for enhancing health outcomes, since it is unlikely that improvements in one domain will be reflected in the other. This was corroborated by Saeidifard, Medina-Inojosa [57] who demonstrated that resistance training alone was associated with a 21% reduction in all-cause mortality, while the combination of resistance and aerobic training was associated with a 40% reduction in risk.

Some physical tests were more sensitive to age-related differences than others. Based on the magnitude of the effect size, handgrip strength was the test least affected by age. This may suggest that this widely used test may not capture subtle age-related declines in a sample of healthy adults (Fig 3). Handgrip strength is often used as a measure of overall muscle strength [58]. Although it does decline with age [59], this decline is typically slower in healthy adults, and become more pronounced in the old age [59]. This may explain why handgrip strength is not as sensitive to subtle, early age-related declines in physical performance. Not surprisingly, the variables that showed the greatest sensitivity to age were, in order from most to least sensitive, CRF, sit-to-stand power, and single-leg stance (i.e., balance performance). CRF is strongly influenced by age as the heart, lungs and muscles naturally lose some efficiency over time. In our sample of healthy individuals, we found a decline in VO$_2$max from the 21–30 age group, confirming that CRF is often a strong indicator of age-related physical decline. Lower body strength and power typically decline faster with age than upper body strength. In fact, we found the first major decline in the 31–40 age group. Lower body strength is crucial for maintaining functional independence, and a decline here is often associated with reduced mobility and quality of life. Balance requires the coordination of several systems (e.g., vestibular, proprioceptive and muscular), which can deteriorate with age [60]. Poor balance is often one of the first signs of physical decline in older adults [52] and can lead to an increased risk of falling. This makes it a highly sensitive test for detecting age-related changes, and indeed we found the first decline in the 41–50 age group. Taken together, the present findings confirm that a combination of tests targeting different physiological systems (e.g., cardiovascular, neuromuscular and musculoskeletal) provides a more comprehensive view to fully assess age-related decline.

The PCA found three main components in our dataset. The first component, "Cardiorespiratory fitness" was composed by the estimated VO$_2$max and flexibility. While it is not surprising that CRF is the most important factor, the connection between VO$_2$max (which depends on muscle oxygen consumption and cardiovascular fitness) and flexibility (which involves joint mobility and muscle extensibility) might seem less intuitive. Such a finding can be explained by the fact that individuals who engage in regular physical activity aimed at improving CRF (e.g., running, cycling, swimming) might also incorporate stretching or flexibility training as part of their routine or tend to have better flexibility as well. In any case, this finding suggests that the inclusion of a VO$_2$max estimates is necessary when the aim is to detect age-related or fitness-related differences in a sample of apparently healthy individuals. The second components, "strength and power", regards the two outcomes mostly related to musculoskeletal fitness, i.e., handgrip strength and sit-to-stand power. The two tests focus separately on upper and lower limb functions, and they both depend on muscle mass quantity and quality, such as fat infiltration, loss of type II fibers or motor units, which typically occur with ageing. Both strength outcomes have been linked to health and longevity, and therefore it is reasonable that they belong to the same component. The last component

is the sensorimotor function, and it is an underrated one in the context of fitness assessment. It is composed by finger tapping test and single-leg stance performance (even if this variable is weakly represented by the component). The finger tapping test provides an estimate of hand dexterity, which tends to decrease with age as can be seen from 50 years of age on in our sample (Fig 2). The same is valid for balance performance which greatly increased in the same age groups (Fig 2) and which increase has been linked to higher risk of falls in adults older than 50 years [18,19]. The three components of the PCA showed different sensitivity to age differences, with "Cardiorespiratory fitness" being the most sensitive, followed by "Strength and power" and by "Sensorimotor function" (Fig 4). The average of the three components also showed large sensitivity to age differences, with the 41–50 age group being the first to be statistically significant different from the 18–20 age group (Fig 5).

One of the aims of performing physical tests on apparently healthy individuals is to raise awareness about their current physical status. When presenting individual results, there are two seemingly contradictory needs to consider. First, people need personalized feedbacks: providing participants with specific insights into their own strengths and weaknesses is essential. Personalized advice is more effective than general recommendations because it directly targets each individual's unique characteristics [61,62]. This allows participants to focus on areas to be improved and capitalize on their strengths, making the advice more relevant and actionable for them [63]. At the same time, there is a need to give participants a simple, immediate score that reflects their overall fitness. This score serves as a benchmark for individuals to easily understand their general health and fitness status. It can motivate them to improve their physical condition through increased physical activity or lifestyle changes. Balancing these two needs ensures that participants receive both detailed, tailored advice that helps them address specific areas, as well as a clear, overall metric to gauge their progress over time. For these reasons, we presented participants with the z-scores of each test in a radar plot, as shown in Fig 6, so that they could see their best and least developed physical capacities. We also calculated a simple composite score, the PiC score, which is the average of the component scores obtained by the PCA and is able to detect subtle changes in physical performance over time, as can be seen in Fig 5.

As a limitation we acknowledge that we recruited a higher proportion of female participants and this might have potential impact on absolute performance values, particularly for strength- and power-related outcomes. However, all analyses targeting age-related differences and multivariate structure were conducted on sex-standardized z-scores, which mitigates this imbalance by removing systematic sex-related differences. From a practical perspective, the PiC Score should be interpreted as a multidomain indicator of physical capacity: lower values reflect concurrent reductions across physiological systems that underpin everyday function. Importantly, because the PiC Score is expressed in standardized units derived from sex-specific normalization and PCA components, it is intended to facilitate comparison across domains within individuals and across age groups rather than to provide an immediate clinical risk estimate. Translating a given change (e.g., 1 SD) into absolute functional impairment or health risk will require clinical anchoring and longitudinal follow-up studies linking the PiC Score to objective outcomes such as incident falls, loss of independence, hospitalization, and mortality. Finally, the test–retest subsample was relatively small (n = 24), which may reduce the precision of reliability estimates. Thus, reliability analysis should be interpreted as preliminary and confirmed in larger independent samples.

In conclusion, we suggest that this developed platform may be a useful tool for assessing physical function and identifying individuals at risk of adverse health outcomes. The low correlations between outcomes showed that the adopted physical tests captured distinct aspects of physical capacity highlighting the multidimensional nature of physical function. This finding underlined the importance of using a comprehensive battery of tests to gain a holistic understanding of an individual's physical health. The platform's ability to provide detailed and specific insights has important implications for tailoring interventions aimed at improving physical health and performance across various populations. Future studies should clinically validate the proposed score and employ longitudinal designs to determine its ability to predict long-term health outcomes, including functional decline and longevity.

## Supporting information

**S1 Appendix. Bland-Altman plots for each test and subject.**
(DOCX)

## Author contributions

**Conceptualization:** Gennaro Boccia, Paolo Riccardo Brustio, Anna Mulasso, Alberto Rainoldi.

**Data curation:** Gennaro Boccia, Paolo Riccardo Brustio, Francesco Tufo.

**Formal analysis:** Gennaro Boccia, Paolo Riccardo Brustio.

**Funding acquisition:** Alberto Rainoldi.

**Investigation:** Gennaro Boccia, Paolo Riccardo Brustio, Anna Mulasso, Francesco Tufo.

**Methodology:** Gennaro Boccia, Paolo Riccardo Brustio, Anna Mulasso, Francesco Tufo, Alberto Rainoldi.

**Project administration:** Alberto Rainoldi.

**Resources:** Anna Mulasso.

**Software:** Gennaro Boccia.

**Supervision:** Alberto Rainoldi.

**Validation:** Gennaro Boccia, Paolo Riccardo Brustio.

**Visualization:** Gennaro Boccia, Anna Mulasso, Francesco Tufo.

**Writing – original draft:** Gennaro Boccia, Paolo Riccardo Brustio, Anna Mulasso, Francesco Tufo.

**Writing – review & editing:** Gennaro Boccia, Paolo Riccardo Brustio, Anna Mulasso, Francesco Tufo, Alberto Rainoldi.

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
