## [Decision Letter · Decision Letter 0]

22 Nov 2025

Dear Dr. Mulasso,

Thank you for submitting your manuscript to PLOS ONE. After careful consideration, we feel that it has merit but does not fully meet PLOS ONE’s publication criteria as it currently stands. Therefore, we invite you to submit a revised version of the manuscript that addresses the minor points raised during the review process (see comments attached).

We look forward to receiving your revised manuscript.

Kind regards,

Laurent Mourot

Section Editor

PLOS ONE

Journal Requirements:

Reviewer's Responses to Questions

**Comments to the Author**

1. Is the manuscript technically sound, and do the data support the conclusions?

Reviewer #1: Yes

2. Has the statistical analysis been performed appropriately and rigorously?

Reviewer #1: Yes

3. Have the authors made all data underlying the findings in their manuscript fully available?

Reviewer #1: Yes

4. Is the manuscript presented in an intelligible fashion and written in standard English?

Reviewer #1: Yes

Reviewer #1: This is a well-conceived, methodologically rigorous, and clearly written manuscript.

The authors present an innovative and comprehensive platform for assessing physical capacities using validated tests integrated through a custom-built hardware and software system.

The study is original, statistically sound, and relevant to research on physical fitness, aging, and health monitoring.

I consider the paper ready for acceptance after minor revisions.

Major Strengths

1. Clarity and coherence: The manuscript is logically structured and easy to follow, with a clear link between the research rationale, methodology, and conclusions.

2. Strong methodological foundation: The study uses a large sample (n = 812), well-established physical tests, and appropriate analytical methods (ICC, COV, ANOVA, PCA).

3. Innovation and applicability: The development of a stand-alone, automated platform represents an important technological advance with high translational potential.

4. Comprehensive scope: The inclusion of multiple domains of physical fitness (strength, flexibility, balance, dexterity, and cardiorespiratory fitness) provides a holistic assessment of physical capacity.

5. Scientific relevance: The findings offer meaningful insights into age-related physical decline and support the need for multifaceted assessment tools.

Suggestions for Minor Improvement

1. Clarification of the PiC Score computation: The manuscript would benefit from a clearer description of how the composite PiC Score is derived. Please specify how z-scores and PCA components are integrated, and whether weighting factors were applied. This would improve methodological transparency and reproducibility.

2. Sample composition: The sample is predominantly female (63.5%). Please discuss briefly whether this gender imbalance might affect test outcomes (e.g., strength or power) and whether normalization by sex sufficiently mitigates this bias.

3. Test–retest reliability subsample: The test–retest analysis includes only 24 participants. A brief justification of this subsample size or acknowledgment of its limitation would strengthen the reliability section.

4. Practical interpretation of age-related differences: Consider expanding the discussion to contextualize what the observed age-related differences represent in practical or functional terms, for example, how a 1 SD decline in the PiC Score translates into meaningful changes in physical function or health risk.

5. Figures and legends: Figures are clear and informative. However, adding brief notes in the legends to clarify statistical symbols (p < 0.05, p < 0.001) and to state that values are expressed as standardized z-scores would make them more self-contained.

6. Conclusion and future applications: The conclusion is concise and appropriate, but it could be slightly enhanced by mentioning potential next steps, such as clinical validation, digital health integration, or longitudinal follow-up studies.

**Do you want your identity to be public for this peer review?** For information about this choice, including consent withdrawal, please see our Privacy Policy

Reviewer #1: **Yes:** Carolina Alexandra Cabo

---

## [Author Response · Author response to Decision Letter 1]

19 Jan 2026

Response to Reviewer #1

We sincerely thank the Reviewer for the careful evaluation of our manuscript and for the positive and constructive feedback. We are pleased that the Reviewer found the study to be methodologically rigorous, clearly written, and relevant. Below, we address each comment point by point. Reviewer comments are reported verbatim, followed by our responses and a description of the revisions made to the manuscript.

General Comment

Reviewer #1:

This is a well-conceived, methodologically rigorous, and clearly written manuscript. The authors present an innovative and comprehensive platform for assessing physical capacities using validated tests integrated through a custom-built hardware and software system. The study is original, statistically sound, and relevant to research on physical fitness, aging, and health monitoring. I consider the paper ready for acceptance after minor revisions.

Response:

We thank the Reviewer for this very positive overall evaluation of our work. We have carefully addressed all the suggested minor revisions to further improve clarity, transparency, and practical relevance of the manuscript.

Suggestions for Minor Improvement

Comment 1

Clarification of the PiC Score computation: The manuscript would benefit from a clearer description of how the composite PiC Score is derived. Please specify how z-scores and PCA components are integrated, and whether weighting factors were applied. This would improve methodological transparency and reproducibility.

Response:

We fully agree with the Reviewer that a clearer description of the PiC Score computation is essential for transparency and reproducibility. We have therefore expanded the relevant sections of the Methods and Discussion to explicitly describe the computation steps. Specifically, we clarified that all test outcomes were first standardized by sex (z-scores), that PCA was conducted on these standardized variables, and that the PiC Score was computed as the unweighted average of the PCA component scores. We also explicitly stated that no weighting factors were applied to individual components. These clarifications have been added in statistical analysis section to improve methodological clarity.

Comment 2

Sample composition: The sample is predominantly female (63.5%). Please discuss briefly whether this gender imbalance might affect test outcomes (e.g., strength or power) and whether normalization by sex sufficiently mitigates this bias.

Response:

We agree with the Reviewer that the gender imbalance in the sample deserves explicit discussion. We have added a paragraph in the Discussion acknowledging the higher proportion of female participants and its potential impact on absolute performance values, particularly for strength- and power-related outcomes. We further clarified that all analyses targeting age-related differences and multivariate structure were conducted on sex-standardized z-scores, which mitigates this imbalance by removing systematic sex-related differences. This limitation and the rationale for sex normalization are now explicitly addressed.

Comment 3

Test–retest reliability subsample: The test–retest analysis includes only 24 participants. A brief justification of this subsample size or acknowledgment of its limitation would strengthen the reliability section.

Response:

We thank the Reviewer for this suggestion and agree that the sample size for the test–retest analysis should be contextualized. We have added a brief justification in the Methods and Discussion, noting that similar sample sizes are commonly used in reliability studies involving repeated laboratory-based assessments.

Comment 4

Practical interpretation of age-related differences: Consider expanding the discussion to contextualize what the observed age-related differences represent in practical or functional terms, for example, how a 1 SD decline in the PiC Score translates into meaningful changes in physical function or health risk.

Response:

We agree with the Reviewer that practical interpretation strengthens the translational value of the manuscript. We have expanded the Discussion to contextualize the observed age-related differences in functional terms by linking decrements in the PiC Score to changes across multiple domains that are directly relevant to everyday function (cardiorespiratory fitness, lower-limb power, balance, dexterity, flexibility, and strength). Because the present study is cross-sectional and the PiC Score is expressed in standardized units (sex-specific z-scores and PCA-derived component scores), we refrain from quantifying how a 1 SD change translates into specific health risks or clinical thresholds, which would require external clinical anchoring and longitudinal outcome data. We now explicitly state this limitation and outline longitudinal validation as a necessary next step to map score changes to clinically meaningful endpoints (e.g., functional decline, falls, cardiometabolic outcomes, and mortality).

Comment 5

Figures and legends: Figures are clear and informative. However, adding brief notes in the legends to clarify statistical symbols (p < 0.05, p < 0.001) and to state that values are expressed as standardized z-scores would make them more self-contained.

Response:

We agree with this suggestion and have revised all relevant figure legends accordingly. Specifically, we double-checked the inclusion of a brief explanations of the statistical symbols used and explicitly stated when values are expressed as standardized z-scores.

Comment 6

Conclusion and future applications: The conclusion is concise and appropriate, but it could be slightly enhanced by mentioning potential next steps, such as clinical validation, digital health integration, or longitudinal follow-up studies.

Response:

We agree with the Reviewer and have slightly expanded the Conclusion to highlight future perspectives. In particular, we now mention potential next steps including clinical validation in patient populations, integration with digital health and preventive screening frameworks, and longitudinal studies aimed at tracking changes in physical capacity over time. This addition better positions the platform within future research and applied contexts.

We once again thank the Reviewer for the constructive feedback, which has helped us improve the clarity, robustness, and practical relevance of the manuscript.

---

## [Editor Report · Decision Letter 1]

2 Feb 2026

Development and validation of the Physical Capacity Score (PiC) to overcome the lack of correlation among traditional physical tests in detecting age-related decline

PONE-D-25-42580R1

Dear Dr. Mulasso,

We’re pleased to inform you that your manuscript has been judged scientifically suitable for publication and will be formally accepted for publication once it meets all outstanding technical requirements.

Kind regards,

Laurent Mourot

Section Editor

PLOS One
---

## [Editor Report · Acceptance letter]

PONE-D-25-42580R1

PLOS One

Dear Dr. Mulasso,

I'm pleased to inform you that your manuscript has been deemed suitable for publication in PLOS One. Congratulations! Your manuscript is now being handed over to our production team.

Kind regards,

on behalf of

Pr Laurent Mourot

Section Editor

PLOS One